# Optimal Shape Design of Direct-Drive Permanent Magnet Generator for 1 kW-Class Wind Turbines

Hyung Joon Park [1], Hyo Lim Kang [1], Dae Gyun Ahn [2] and Seung Ho Han [1,*]

1   Department of Mechanical Engineering, Dong-A University, Busan 49315, Republic of Korea;
    gudwns3481@gmail.com (H.J.P.); hlk12@donga.ac.kr (H.L.K.)
2   Wins, Inc., Haman-gun 52057, Republic of Korea
*   Correspondence: shhan85@dau.ac.kr

**Abstract:** Direct-drive permanent magnet generators are becoming an attractive option for highly efficient small-scale wind turbines due to their high-power density and size reduction capabilities. In this study, the optimal shape design of a direct-drive permanent magnet generator for 1 kW-class wind turbines was conducted while considering power generation and weight. Half of the geometry of a single stage in the generator was considered for a electromagnetic analysis under given electrical parameters. In order to construct a response surface model, a sensitivity analysis was conducted on seven design parameters of the proposed generator. The desirability function was used to minimize the weight of the generator while meeting a requirement of the target specification. The results indicated that the optimized design parameters for the generator met the target specification while maintaining the generator's weight at the same level as the initial design model. From the comparisons with other research, the optimized generator exhibited a higher power generation/weight ratio than the generator with a rated capacity under 3 kW.

**Keywords:** desirability function; direct-drive permanent magnet generator; electromagnetic analysis; response surface method; shape optimal design; sensitivity analysis

## 1. Introduction

Wind turbines are highly effective devices that convert the kinetic energy of wind into mechanical and eventually electrical energy. They are commonly categorized into two types based on their rated capacities and intended applications. Large-scale wind turbines have rated capacities ranging up to several MWs and are designed for use in power plants that are typically deployed in wind farms. In contrast, small-scale wind turbines can be grid-connected for residential or industrial electricity generation or can be used in off-grid applications such as water pumping or battery charging. Small-scale wind turbines are typically installed as a single unit or in small numbers [1]. In particular, wind turbines with a rated capacity of 1 kW have gained attention for their excellent adaptability to the urban environment in terms of lower manufacturing and maintenance costs [2,3].

The 1 kW-class wind turbine utilizes a vertical-axis generator with Darrieus-type blades to facilitate operation even under highly turbulent and irregular wind conditions [4]. To ensure high-efficiency power generation even under weak wind conditions, a direct-drive generator using the axial-flux permanent magnet (AFPM) has been adopted to reduce additional mechanical and electrical losses during power generation. This design allows for operation under rapidly changing wind speed conditions without the use of gearboxes [5]. Moreover, the rated capacity of the generator can be easily adjusted by stacking disk-shaped compact features in the axial-direction [6]. The use of a permanent magnet in the rotor eliminates the need for a field current, which is supplied to the field windings of the generator, resulting in a simple structure and low costs.

However, the disadvantages of AFPM machines have been reported in comparison to other applications, such as a lower torque/mass ratio, large outer diameter, cogging

torque and power losses [7]. Much research regarding cogging torque and electrical losses, which reduce the performance of the machine, have been carried out in the past years to define strategies in the design stage of machines [8]. Arand [9] investigated an effective cogging torque mitigation method for AFPM machines with a yokeless and segmented armature. By implementing the response surface methodology, optimal design factors are determined to mitigate the cogging torque. Hosseini et al. [10] designed and analyzed the performance of an AFPM generator with a non-ferromagnetic permanent magnet holder on the rotor disc. The generator's performance and efficiency were evaluated analytically and experimentally, considering power loss. The results showed an improvement in performance even at high rotating speeds without an increase in the number of pole pairs. Lee et al. [6] demonstrated the reduction of the no-load momentum of the AFPM generator using skewed magnets. This approach provided a significant advantage for the AFPM generator in terms of low cogging torque, high efficiency and compact size. Hüner [11] used the Taguchi experimental method for the optimal design of the AFPM generator. A sensitivity analysis based on the electromagnetic analysis results was conducted to identify optimum values for six independent design parameters and five levels. As a result, the degree of influence of the design parameters on the performance of the generator was determined by a regression model.

The preceding literature reviews elucidated the design approaches for reducing cogging torque and power losses to increase efficiency and output power of the AFPM generator. However, the weight reduction of the generator should be considered for the generator design, since the performance of the AFPM machine's efficiency is influenced not only by the electromagnetic characteristics, but also by the weight and shape of mechanical parts of the generator, leading to excessive rotational momentum and friction losses in the stator and rotor [12,13]. Additionally, the weight of the generator is closely linked to its material cost, which is an important criterion for the wind turbine design [14,15].

There is research on the weight reduction of the AFPM generator. Hendrik et al. [14] investigated the influence of the weight on the efficiency and material cost of the AFPM generator considering four shape design parameters. The electromagnetic analysis was conducted to evaluate the performance of the generator, and the obtained results showed that the weight was significantly reduced with only a small reduction in the efficiency of generator. However, in this case, the shape design parameters were considered on a limited basis. Santiago et al. [15] conducted a dimensioning optimization of the AFPM generator. This methodology allowed optimizing the generator design by establishing useful dimensioning options for a range of magnet radiuses and lengths. The application of this methodology demonstrated the viability of maximizing the efficiency and reducing the weight of the generator. However, the optimization was conducted by considering only a few shape design parameters that influenced the weight and performance of the generator, so it is necessary to identify additional design parameters that impact the performance of the generator.

In this context, an optimal shape design for the AFPM generator is presented to minimize the weight of the generator under the constraint of the required target specification such as an active power. An outer-rotor type generator was proposed, which was designed for a rated capacity of 1 kW. The electromagnetic analysis based on the finite element method was carried out to evaluate the active power of the generator. To investigate the sensitivity of the shape design parameters that impacted the active power and weight of the proposed generator, response surface methodology (RSM) was employed based on the electromagnetic analysis results. Then, the optimal shape design parameters were determined using a desirability function via the obtained response's surface model and compared with other research.

## 2. Outer-Rotor Type Generator with AFPM

### 2.1. Model Descriptions

Figure 1a shows the configuration of a vertical-axis wind turbine (VAWT) with three Darrieus-type blades attached to a rotating vertical shaft. The lengths of each blade and reaction arm are 2 m and 1.3 m, respectively, and the weight of the blades including reaction arms is 17.7 kgf. The rated rotation speed of the vertical shaft is 100 rpm at a rated wind speed of 10 m/s, and the sweep area of the blades is 5.2 m². The rated power of the VAWT was estimated based on both the operation conditions and the sweep area, using a power coefficient of 0.35 for a general Darrieus-type wind turbine [16]. Eventually, the rated power was designated as 1 kW. To generate electricity at the rated power of 1 kW, an outer-rotor type generator with AFPM was proposed, as shown in Figure 1b. The rotor, which is supported by bearings, has permanent magnets mounted up and down evenly, while the stator, wound by winding coils, is stationary. This feature enables the occurrence of axial-flux in the generator, which can be stacked vertically with two stages in the center of the tower. The requirement of power generation per stage was 500 W, with a torque of 48 Nm applied to the rotating vertical shaft under operation conditions.

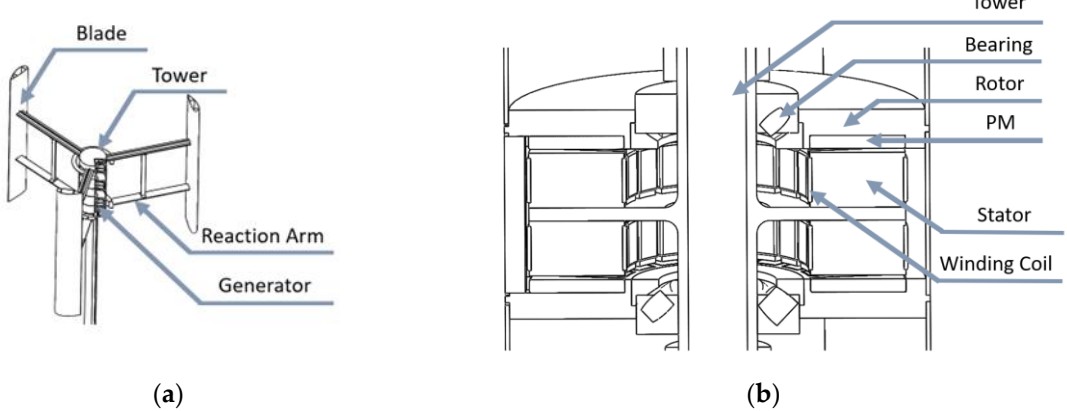

(**a**)　　　　　　　　　　　　　　　　　(**b**)

**Figure 1.** Configuration of vertical-axis wind turbine (VAWT) and outer-rotor type generator with axial-flux permanent magnet (AFPM): (**a**) VAWT; (**b**) AFPM generator.

A slotted stator core was laminated radially by 100 stacks of ferromagnetic steel plates with a thickness of 0.8 mm, as shown in Figure 2. This type of core allows for a smaller air gap between the stator core and permanent magnet and can obtain higher magnetic flux density compared to the slotless type stator.

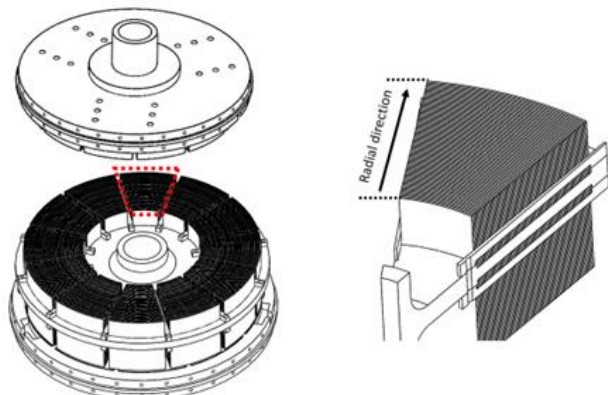

**Figure 2.** Spiral lamination of slotted stator core.

The design of the winding coils, along with the combination of poles and slots, is one of the important factors determining the performance of the generator. There are two winding

methods for the AFPM generator based on the winding shape and coil pitch: concentrated and distributed winding. In this study, the distributed winding method was used, which has been known to provide excellent harmonic disturbance rejection, leakage reactance and thermal dynamic characteristics [17,18]. Additionally, a two-layer method for winding was considered to contain an even number of coil-sides in two layers. Figure 3 shows the arrangement of stator windings for three phases: A, B and C. To meet the requirements for low rotational speed and a winding factor value of 0.933, the combination of 12-slot and 10-pole were adopted [19].

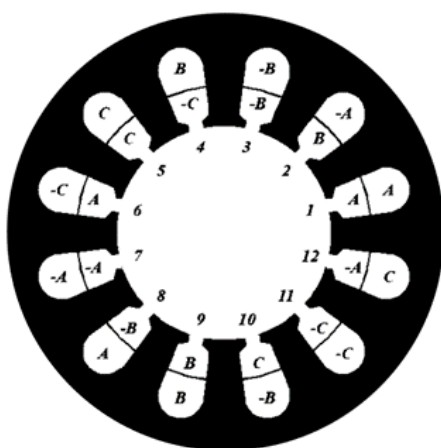

**Figure 3.** Arrangement of stator windings.

### 2.2. Shape Parameters

At the initial design stage of the outer-rotor type generator with the AFPM, various shape parameters were considered as shown in Figure 4. The inner diameter and length of the stator were denoted by $D_i$ and $l$, respectively, as shown in Figure 4a. The stator slot had detailed shape parameters, including the slot opening $b_1$, slot width $b_2$, opening height $h_1$ and slot height $h_2$, which were represented by a cross-section view of AA'BB' as shown in Figure 4b. The shape parameters of the permanent magnets were the same as those of the stator, such as $D_i$ and $l$, while the thickness of the permanent magnets was 10 mm and the distance between adjacent permanent magnets was 3 mm. The air gap clearance between the stator and permanent magnet was set at 1 mm, taking into account the manufacturing cost and maintenance aspects. The initial design stage specifications of the shape parameters and the number of winding coil turns $N_{ph}$ are summarized in Table 1. Furthermore, the materials used at the initial design stage are listed in Table 2.

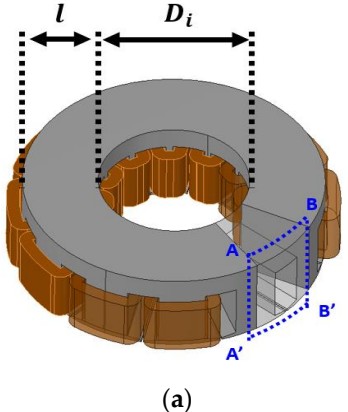

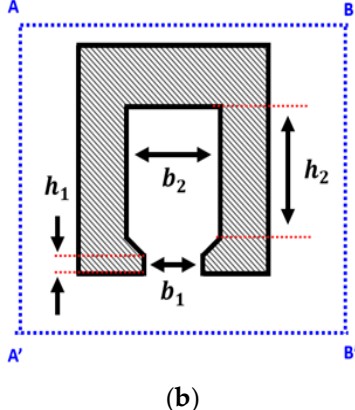

(**a**)  (**b**)

**Figure 4.** Shape parameters of the generator: (**a**) Parameters for outside of stator; (**b**) Parameters for inside of stator at cross-section area of AA'BB'.

**Table 1.** Specification of shape parameters and the number of winding coil turns at initial design stage of the generator.

| Parameters | Magnet Thickness | Air Gap | $D_i$ | $l$ | $b_1$ | $b_2$ | $h_1$ | $h_2$ | $N_{ph}$ |
|---|---|---|---|---|---|---|---|---|---|
| Values | 10 | 1 | 65 | 35 | 2 | 8 | 2 | 16 | 200 |
| Unit | | | | mm | | | | | Turn |

**Table 2.** Materials used in the generator.

| Parts | Materials |
|---|---|
| Permanent Magnet | NdFe30 |
| Stator | D21-50 |
| Bearing | Stainless steel |
| Coil | Copper |
| Rotor | D21-50 |

## 3. Electromagnetic Analysis

An electromagnetic analysis was conducted to evaluate the electromagnetic characteristics of the generator constructed at the initial design stage. To reduce the computational time consumption required in the electromagnetic analysis using a three-dimensional model of the AFPM generator, a two-dimensional model approach was used [20]. The principle of transforming the three-dimensional model to a corresponding two-dimensional model is illustrated in Figure 5, where a reference plane was selected on the average radius of the AFPM generator. On the selected reference plane, the three-dimensional model can be simplified as a two-dimensional linear generator. Using the two-dimensional model, the flux linkage with induced electromotive force was calculated, and the induced current and voltage were evaluated under the rated-load condition via the electromagnetic analysis. Finally, the output power was obtained using the power formulation and voltage equation.

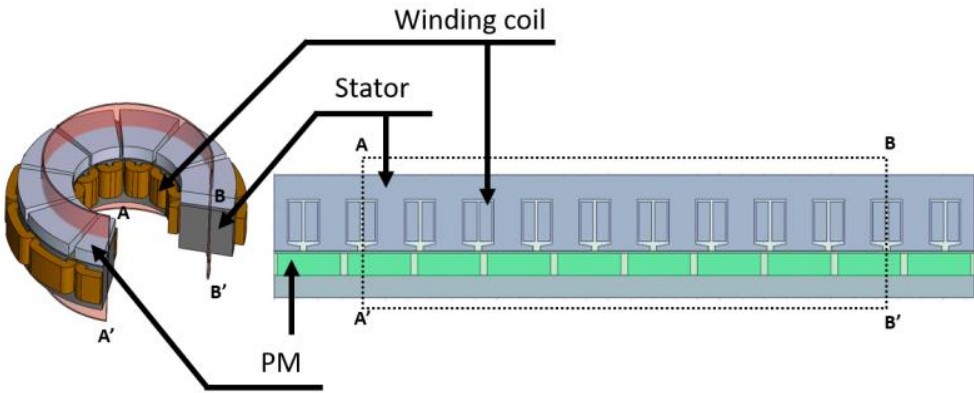

**Figure 5.** A principle how to transform the 3-D geometry of an AFPM generator to a 2-D geometry.

### 3.1. Magnetic Flux Linkage and Induced Electromotive Force

The initial design model of the generator was evaluated for its electromagnetic characteristics using commercial numerical analysis software: ANSYS RM-Expert and Maxwell [21]. The governing equations for the magnetic fields used in the numerical analysis are presented by Equation (1) to (3). Equation (1) represents Ampere's law for the electric current and magnetic field, while Equation (2) shows the electric potential based on Faraday's law and the relationship between electric and magnetic fields. Gauss's law for a magnetic field is expressed by Equation (3) [22]:

$$\nabla \times \mathrm{H} = \mathrm{J} \tag{1}$$

$$\nabla \times E_e = -\frac{\partial}{\partial t} B \tag{2}$$

$$\nabla \cdot B = 0 \tag{3}$$

where $B = (\nabla \cdot A)$, H is magnetic field strength (A/m), J is the current density (A/m$^2$), t is time (s), $E_e$ is Maxwell's electric field, A is the magnetic vector potential and B is the magnetic flux density field.

Figure 6 presents the mesh generation scheme used for the two-dimensional electromagnetic analysis of the AFPM generator. As the generator possesses a symmetrical feature, a half model was utilized to expedite the analysis. To balance computational efficiency and simulation accuracy, the mesh generation was optimized through a trade-off between the mesh density and the computation time. Fine meshes tend to increase simulation time, while coarser meshes yield poorly resolved outputs. Therefore, length-based meshes were adopted with a mesh size of 0.5 mm for the air-gap and coarse meshes with a larger size of 3 mm for the remaining regions, to ensure both computational efficiency and simulation accuracy [23].

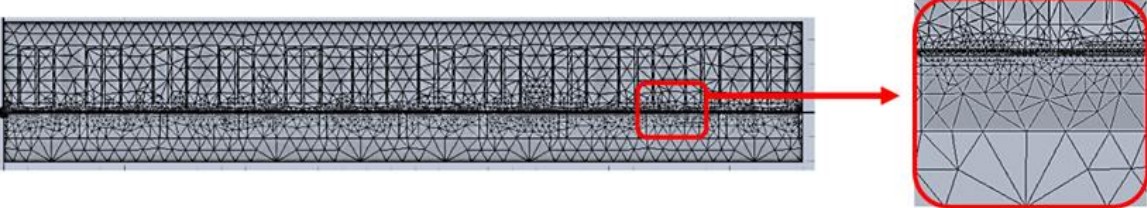

**Figure 6.** Mesh generation of two-dimensional finite element model.

To evaluate the generator's performance, an electromagnetic analysis under no-load conditions was conducted to investigate the flux linkage and electromotive force. The obtained results are shown in Figure 7a, which provides the waveforms of flux linkage and electromotive force $E_F$ for the three phases, A, B and C, with a peak value of 0.32 Wb and a maximum value of 15.7 V. Figure 7b presents the three-dimensional analysis results for the proposed generator model, which reveals a good agreement of the flux linkage and electromotive force obtained from the two- and three-dimensional models.

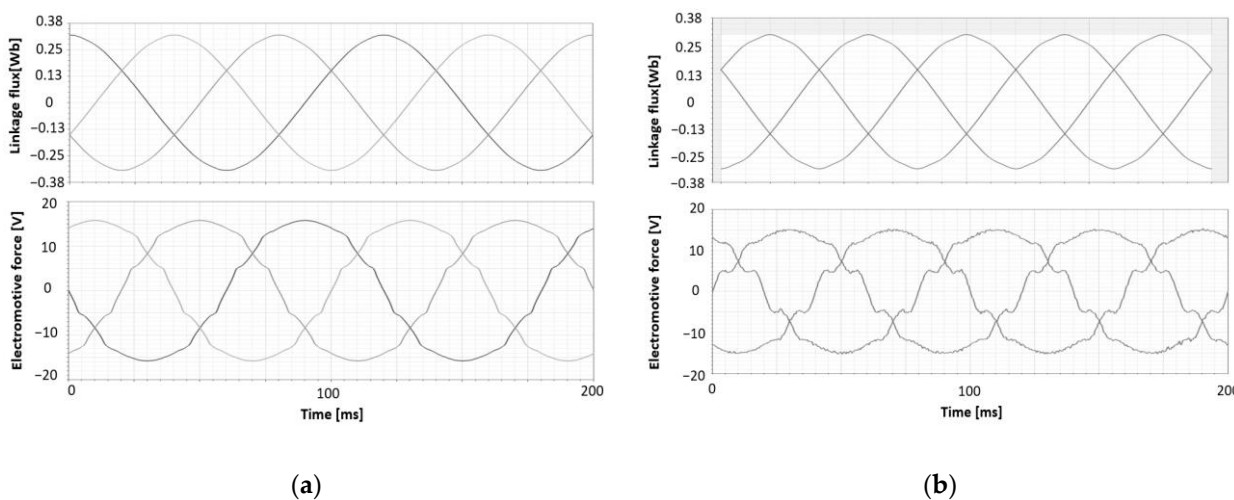

**(a)** **(b)**

**Figure 7.** Flux linkage and electromotive force for A, B and C phases under no-load condition by lapse of time: (**a**) two-dimensional finite element model; (**b**) three-dimensional finite element model.

### 3.2. Equivalent Circuit

To estimate the electrical power output of the AFPM generator, the equivalent circuit and voltage equation must be employed [24]. The circuit and equation include electrical

parameters, such as the phase internal resistance and synchronous inductance. The internal resistance $R_{ph}$ of each phase can be determined using Equation (4), which accounts for the number of turns per phase $N_{ph}$, the average length of a turn $l_w$, the cross-sectional area of the wire for the winding $A_c$ and the electrical conductivity $\rho_c$ of the winding material at a given temperature (for copper, $\rho_c \approx 57 \times 10^6$ S/m at 20 °C) [25].

$$R_{ph} = N_{ph}\rho_c\frac{l_w}{A_c} \tag{4}$$

The synchronous inductance $L_{ph}$ for each phase can be computed by calculating the linkage flux generated by the induced current in the winding coil [25]. When using NdFe-based permanent magnets, the synchronous inductances of the A, B and C phases are equal, since the relative recoil permeability becomes 1.0 [26]. The following equation represents the synchronous inductance:

$$L_{ph} = m\mu_0\frac{1}{\pi}\left(\frac{N_{ph}k_{wn}}{p}\right)^2\frac{\left((D_i + 2l)^2 - D_i^2\right)}{4g} \tag{5}$$

where m is the number of winding phases, $\mu_0$ is the permeability of free space ($0.04\pi \times 10^{-6}$ H/m), $N_{ph}$ is the number of turns per phase for winding, $k_{wn}$ is the winding factor, p is the total number of pole pairs, l is the slot length of stator, $D_i$ is the inner diameter and g is the air gap.

To consider the load condition, Figure 8 illustrates the three-phase equivalent circuit diagram. A Y-connection was employed for the three-phase winding coils, and the load resistance was represented by Load-R, which corresponds to the maximum output power [5].

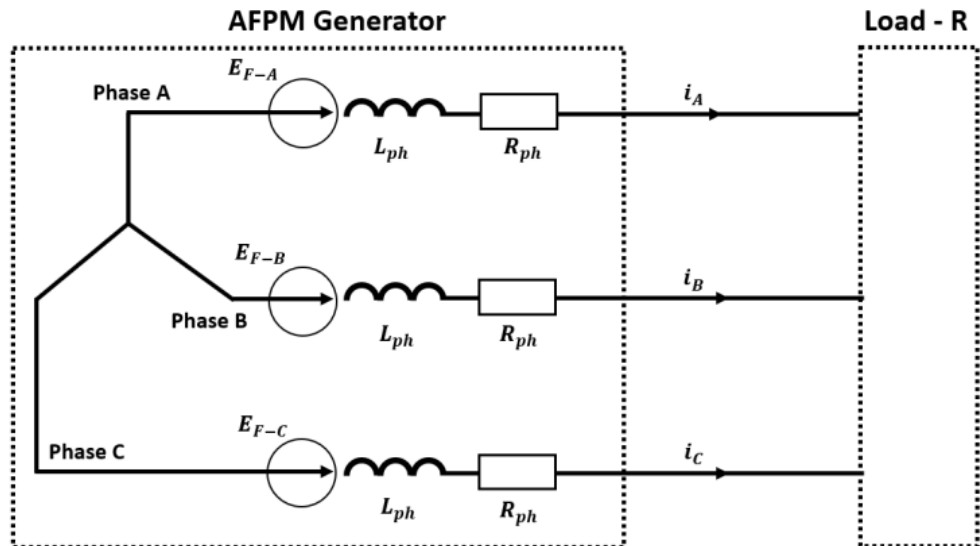

**Figure 8.** Equivalent circuit for three-phase of AFPM generator.

*3.3. Output Power*

For the initial design model of the proposed AFPM generator, the output power under different load conditions can be readily calculated using the power formulation [27]. Equation (6) presents the power formulation, and the voltage equations for each phase can be expressed as Equation (7) [18,24].

$$P = (V_Ai_A + V_Bi_B + V_Ci_C)\cos\delta \tag{6}$$

$$\begin{bmatrix} V_A \\ V_b \\ V_c \end{bmatrix} = \begin{bmatrix} -\left(R_{ph} + R + \frac{d}{dt}L_{ph}\right) & 0 & 0 \\ 0 & -\left(R_{ph} + R + \frac{d}{dt}L_{ph}\right) & 0 \\ 0 & 0 & -\left(R_{ph} + R + \frac{d}{dt}L_{ph}\right) \end{bmatrix} \begin{bmatrix} i_A \\ i_B \\ i_C \end{bmatrix} + \begin{bmatrix} e_A \\ e_B \\ e_C \end{bmatrix} \tag{7}$$

In these equations, P is the output power, $V_A$, $V_B$ and $V_C$ are the induced voltages in the phases of A, B and C for the rated-load condition, $i_A$, $i_B$ and $i_C$ are the induced currents in the phases of A, B and C, and δ presents the phase angle between induced voltages and induced currents. In addition, R is the load resistance and $e_A$, $e_B$ and $e_C$ are back-electromotive forces for each phase.

The back-electromotive forces can be obtained using these equations provided:

$$e_A = -\omega_e \varphi_f \sin(\theta) \tag{8}$$

$$e_B = -\omega_e \varphi_f \sin\left(\theta - \frac{2\pi}{3}\right) \tag{9}$$

$$e_C = -\omega_e \varphi_f \sin\left(\theta + \frac{2\pi}{3}\right) \tag{10}$$

where $\omega_e$ is the electrical angular velocity and $\varphi_f$ is the maximum flux linkage by the permanent magnet.

The waveforms of induced currents and voltages under the rated-load condition are shown in Figure 9, in which the output power can reach a maximum value. The active output power of the proposed AFPM generator was obtained as 199 W, which was corrected by considering the power factor of 0.8. However, the requirement for the active output power is more than 250 W. Therefore, it is necessary to increase the output power by improving the shape parameters, as shown in Figure 4. In the present study, an optimal shape design was carried out to meet the target specification, including the active power of 250 W under a constraint of weight minimization.

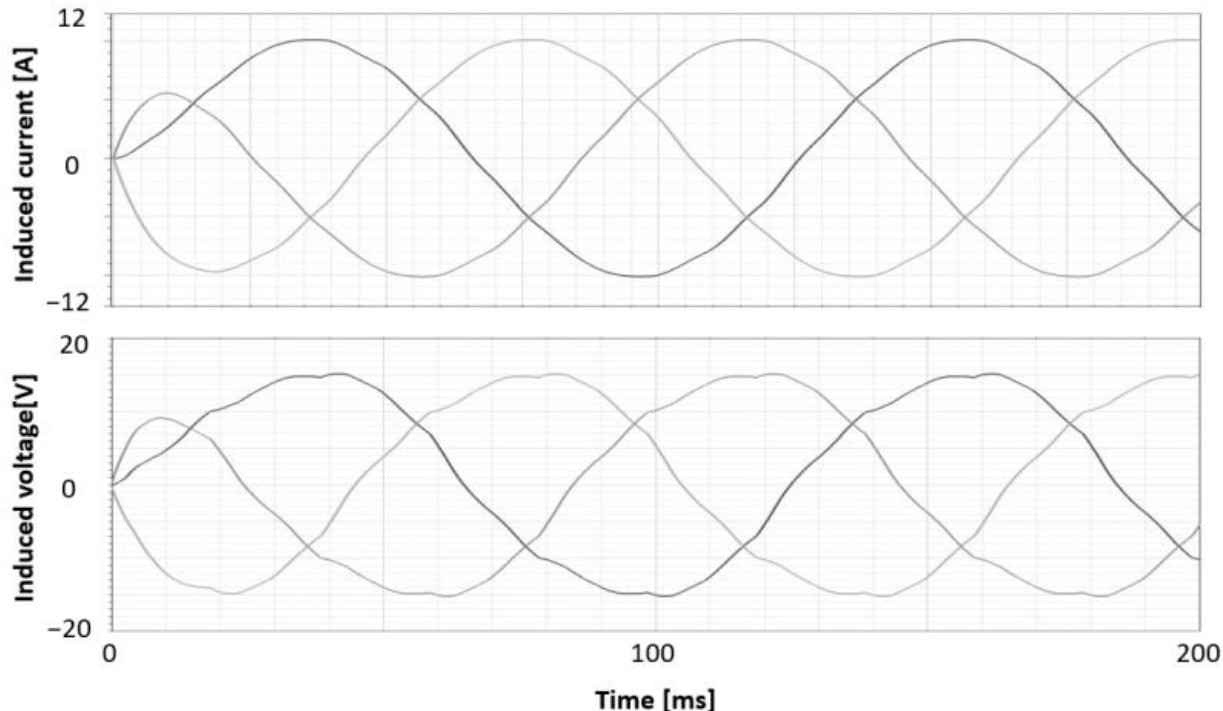

**Figure 9.** Induced currents and voltages for A, B and C phases under rated-load condition by lapse of time.

## 4. Optimal Shape Design

### 4.1. Design Space Exploration

A design space exploration was conducted in order to identify design solutions that satisfy desired design requirements from a space of tentative design points resulting from applying optimizations at various levels of abstraction. To determine the optimal design parameters for the half model of the AFPM generator, as shown in Figure 6, an optimal shape design was carried out with weight W and active power P as the objective function and constraint, respectively.

$$\text{Minimize } W = W\left(D_i, l, b_1, b_2, h_1, h_2, N_{ph}\right)$$
$$\text{Subject to } P\left(D_i, l, b_1, b_2, h_1, h_2, N_{ph}\right) \geq 250[W] \tag{11}$$
$$\text{Design Parameters } D_i, l, b_1, b_2, h_1, h_2, N_{ph}$$

The minimization of weight W and the requirement of active power P greater than or equal to 250 W were considered in this formulation. The optimal shape design consisted of seven design parameters, including six shape parameters such as $D_i, l, b_1, b_2, h_1$ and $h_2$, as shown in Figure 4, and the number of winding coil turns, $N_{ph}$. To select specific parameter values, three levels of abstraction were adopted and specified in Table 3.

**Table 3.** Design parameters specified by 3 levels of abstraction.

| Level | $D_i[mm]$ | $l[mm]$ | $b_1[mm]$ | $b_2[mm]$ | $h_1[mm]$ | $h_2[mm]$ | $N_{ph}[turns]$ |
|-------|-----------|---------|-----------|-----------|-----------|-----------|-----------------|
| 0 | 60 | 30 | 1 | 6 | 1 | 10 | 100 |
| 1 | 70 | 35 | 2 | 10 | 2 | 15 | 200 |
| 2 | 80 | 40 | 3 | 14 | 3 | 20 | 300 |

An orthogonal array was utilized to estimate the effect of different design parameters on performance characteristics in a condensed set of analyses [28]. For this purpose, an appropriate orthogonal array such as L27($3^7$) was selected based on the number of design parameters and levels, indicating that only 27 experiments were required to study the effect of a maximum of seven variables at three levels each, instead of conducting $3^7$ experiments. The 27 experiments for the design space exploration are listed in Table 4. For each experiment, the electromagnetic analysis under the rated-load condition was carried out, in which the active powers were calculated by induced voltages and currents using Equation (7). The resistive load values were considered to maximize active power. The weights, including coil, slot, permanent magnet, back iron disk and outer cover, were estimated by considering the design parameters of each experiment in the orthogonal array.

**Table 4.** Orthogonal array of L27($3^7$) for design space exploration.

| No. | 1 | 2 | 3 | 4 | 5 | $\cdots$ | 23 | 24 | 25 | 26 | 27 |
|-----|---|---|---|---|---|----------|----|----|----|----|----|
| Parameters | | | | | 3 levels (0, 1, 2) | | | | | | |
| $D_i$ | 0 | 0 | 0 | 0 | 0 | | 2 | 2 | 2 | 2 | 2 |
| $l$ | 0 | 1 | 2 | 0 | 1 | | 1 | 2 | 0 | 1 | 2 |
| $b_1$ | 0 | 0 | 0 | 1 | 1 | | 2 | 2 | 0 | 0 | 0 |
| $b_2$ | 0 | 1 | 2 | 0 | 1 | $\cdots$ | 2 | 0 | 1 | 2 | 0 |
| $h_1$ | 0 | 1 | 2 | 1 | 2 | | 1 | 2 | 1 | 2 | 0 |
| $h_2$ | 0 | 0 | 0 | 1 | 1 | | 1 | 1 | 2 | 2 | 2 |
| $N_{ph}$ | 0 | 1 | 2 | 1 | 2 | | 0 | 1 | 0 | 1 | 2 |
| Function | | | | | Analysis result | | | | | | |
| P [W] | 61.8 | 138.7 | 195.5 | 98.4 | 198.4 | | 368.2 | 239.9 | 233 | 283.4 | 300.6 |
| W [kgf] | 3.34 | 4.14 | 5.02 | 3.84 | 4.75 | | 5.69 | 6.6 | 5.31 | 6.5 | 7.08 |

To quantify the variations of performance characteristics regarding changes in design parameters, a sensitivity analysis was conducted [29]. The influence of the design parameters on weight W and active power P was evaluated using a stepwise regression, in which the overall regression correlation coefficient, such as the $R^2$-value, was up to 0.9. The results of the sensitivity analysis are shown in Figure 10. The source term indicated the main or interaction effect, and the LogWorth statistic value (defined as $-\log(p - value)$) was used to measure the level at which the observed between the average criterion values was statistically significant. The LogWorth value greater than 2, indicated by the blue dashed line in these figures, was considered statistically significant for inclusion of a factor at a 1% significance level [30]. Among the design parameters, there was high sensitivity on both the active power and the weight, such as the inner diameter $D_i$, the length of the stator l and slot height $h_2$. These parameters were related to the size of the generator and the effects on electromagnetic properties, such as the magnetic flux excited permanent magnets, permeance coefficient and electromotive force [25]. The slot opening $b_1$ was related to the fundamental frequency of magnetic flux, which showed an effect on the efficiency of the generator, such as losses caused by using ferromagnetic cores for the slotted stator [25]. However, this parameter provided relatively lower sensitivity on the active power and weight. The number of winding turns per phase $N_{ph}$ also showed relatively lower sensitivity on both the active power and weight. However, $N_{ph}$ was able to increase the electromotive force, which could cause an increase in the phase resistance and impose restrictions on maximum power generation when considering load resistance [31]. In addition, this parameter provided lower sensitivity on weight, since the number of windings only increased while the cross-section area of the stator slot was limited by other design parameters such as slot width and slot height. The slot width $b_2$ exhibited a low sensitivity to weight due to the inverse proportionality between the weight of the iron core and winding coils. However, the electromotive force increased with an increase in $b_2$, and the sensitivity increased when interactions with other design parameters occurred. On the other hand, the opening height $h_1$ had low sensitivity to weight and only affected the distance between the permanent magnet and the winding coil, which was related to the permeance value of the slot.

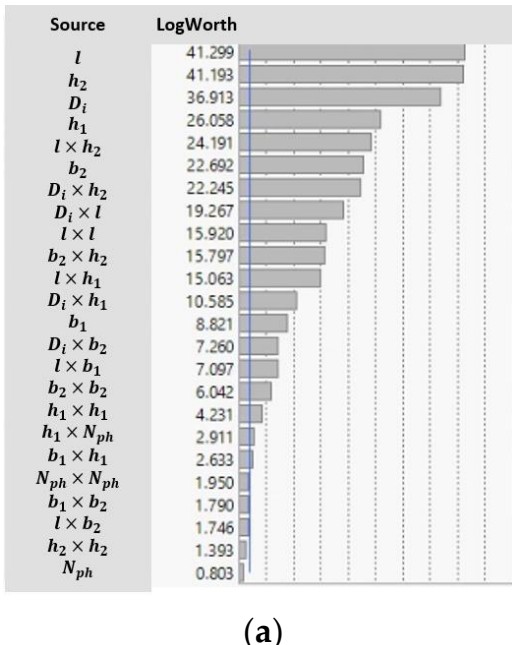
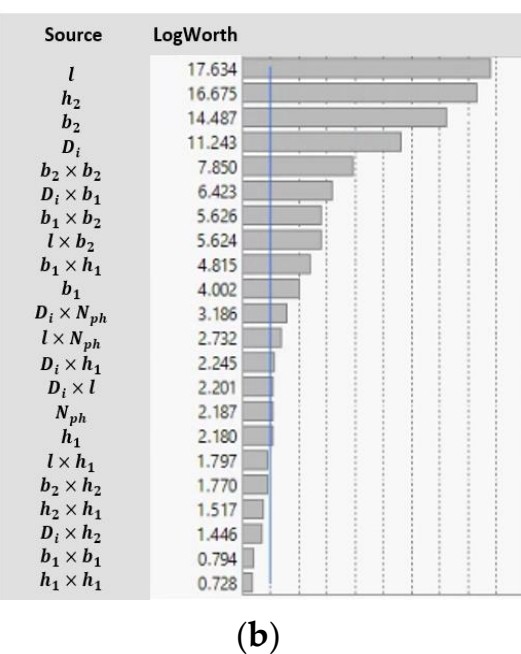

**(a)**      **(b)**

**Figure 10.** Sensitivity analysis for design parameters affecting weight and active power: (**a**) weight W; (**b**) active power P.

In order to establish the functional relationship between weight, active power and design parameters, a statistical analysis method for the response surface was employed. This approach allowed us to represent the changes in responses resulting from complex actions of the design parameters, and their impact on the response variables of the weight and active power. The commercial statistical software JMP [32] was utilized to select the optimal combination of design parameters, while a second-order model was employed in the response surface model [33]. Figure 11 shows several examples of highly sensitive functional relationships between the weight, active power and design parameters. The mathematical model between response variables and design parameters is given below:

$$
\begin{aligned}
\text{Power} = {}& (-907.24) + (3.89 * D_i) + (14.26 * l) + (10.99 * b_1) + (15.50 * b_2) + (14.62 * h_2) + (-6.24 * h_1) \\
& + \left(-0.025 * N_{ph}\right) + (D_i - 68.85) * (l - 35.08) * 0.21 + (D_i - 68.85) * (b_1 - 2.2) * 1.81 \\
& + (D_i - 68.85) * (b_2 - 10.718) * 0.40 + (l - 35.08) * (b_2 - 10.72) * 0.98 + (b_1 - 2.2) \\
& * (b_2 - 10.72) * 3.96 + (b_2 - 10.72)^2 * (-1.88) + (D_i - 68.85) * (h_2 - 14.55) * (-0.13) \cdots
\end{aligned}
\tag{12}
$$

$$
\begin{aligned}
\text{Weight} = {}& (-6.19) + 0.047 * D_i + 0.18 * l + (-0.0055) * b_1 + 0.016 * b_2 + 0.12 * h_2 + 0.097 * h_1 \\
& + (D_i - 68.85) * (l - 35.08)^2 * 0.0012 + (l - 35.08) * (b_1 - 2.2) * (-0.00078) \\
& + (D_i - 68.85) * (b_2 - 10.72) * 0.00036 + (b_2 - 10.72)^2 * (-0.00038) + (D_i - 68.85) \\
& * (h_2 - 14.55) * 0.0012 + (l - 35.08) * (h_2 - 14.55) * 0.0035 \cdots
\end{aligned}
\tag{13}
$$

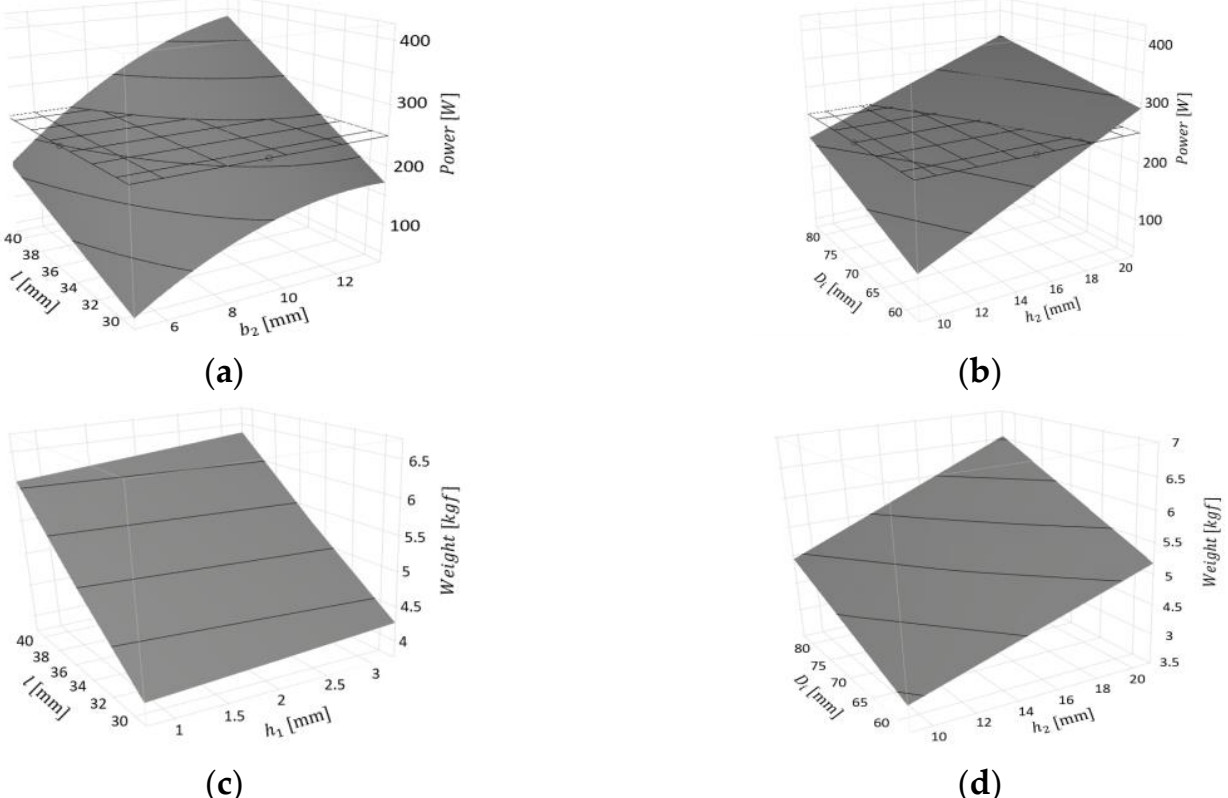

**Figure 11.** Examples of the highly sensitive functional relationship between the weight, active power and design parameters: (**a**) active power by l and $b_2$; (**b**) active power by $D_i$ and $h_2$; (**c**) weight by l and $h_1$; (**d**) weight by $D_i$ and $h_2$.

### 4.2. Optimal Shape Design Using Desirability Function

Through the statistical analysis using the response surface methodology, the optimal design parameters were identified by utilizing the desirability function [34]. The response variables, namely the weight and active power, were transformed into individual desirabil-

ity values $d_j$ for each of the seven design parameters. These values were integrated into the global desirability function $D_s$, which is presented by the following equation:

$$D_s = \left( d_1^{n_1} \times d_2^{n_2} \times \cdots \times d_j^{n_i} \right)^{(1/\sum n_i)} \tag{14}$$

where $n_i$ is the weight-function criteria, set by the designer to determine the relative importance of response variables such as the weight and active power for the generator.

In this study, the individual desirability and weight-function criteria for the weight of the generator were set to 0.8 and 1.0, respectively. For the active power, on the other hand, the individual desirability and weight-function criteria were set to 0.95 and 3.0, respectively, given that it is considered as a constraint condition and hence, more crucial than the weight of the generator.

The impact of seven parameters on response variables can be investigated by using prediction profilers. As shown in Figure 12, the maximized global desirability $D_s$ reached up to a value of 0.83 and the prediction profilers, related to the desirability function, identified slot length l and slot width $b_2$ as the primary factors to optimize the global desirability. However, the desirability function exhibited a decreasing trend when the slot length l exceeded 39.5 mm. This was attributed to the constant desirability for the active power, while the desirability for the weight decreased. The reduction of the inner diameter $D_i$ and slot height $h_2$ was necessary to optimize the global desirability, despite their high sensitivity to both the active power and weight. It was found that these two parameters had lower effects on the response variables than the slot length l. The other parameters were adjusted to maximize the global desirability.

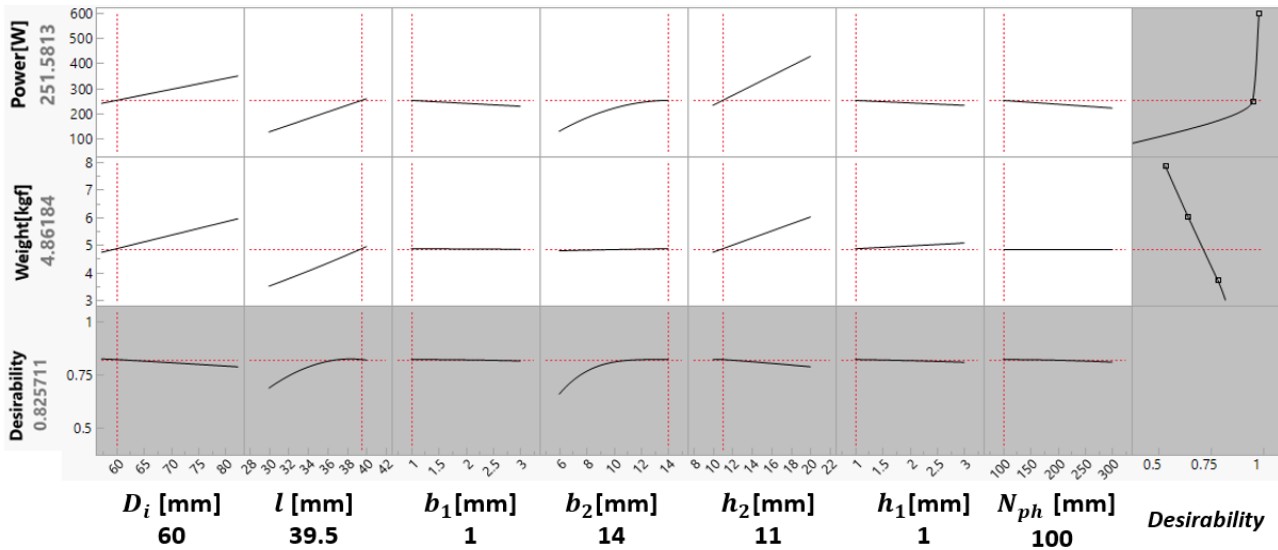

**Figure 12.** Prediction profilers of response variables related to shape parameters.

### 4.3. Result and Discussion

The results obtained from the optimal shape design are presented in Table 5, which includes the results of the initial model as well. The optimal shape design yielded a 25% increase in the active power while maintaining the weight as that of the initial model. In order to validate the results of the optimal shape design, an electromagnetic analysis was performed by using obtained optimized parameters, wherein the circuit and load conditions were taken into account in order to maximize the active power. The results of the electromagnetic analysis for the validation were compared with those obtained by the optimal shape design as shown in Table 6. The error values, which represent the difference of the results from the validation and the optimal shape design, are less than 4%.

**Table 5.** Shape parameters and response variables obtained by shape optimal design.

| Unit | $D_i$ | $l$ | $b_1$ | $b_2$ | $h_1$ | $h_2$ | $N_{ph}$ | $W$ | $P$ |
|---|---|---|---|---|---|---|---|---|---|
| | | | | mm | | | turns | kgf | W |
| Optimized | 60 | 39.5 | 1 | 14 | 1 | 11 | 100 | 4.86 | 251.6 |
| Initial | 65 | 35 | 2 | 8 | 2 | 16 | 200 | 4.99 | 199.2 |

**Table 6.** Validation of results obtained by shape optimal design.

| | Optimized | Validated | Error |
|---|---|---|---|
| W [kgf] | 4.86 | 5.04 | 3.70% |
| P [W] | 251.6 | 252.4 | 0.32% |

The optimal design of the proposed generator successfully satisfied the required active power while maintaining a similar weight difference between the initial and optimal models. This is because the weight-function criteria for the active power in the desirability function was three times higher than that of the weight reduction. If a higher weight-function criteria were imposed on weight minimization, the weight of the generator could be reduced slightly, but the active power of the generator would be lower than the required target specification. Therefore, it is not possible to satisfy the required active power of the generator without increasing its weight under the given shape design parameters. In addition, when determining the shape design parameters, the design parameter for air gap clearance was not considered. The air gap clearance of the proposed generator is 1 mm, which is the same as that of generators with similar specifications [35]. As the air gap clearance decreases by 0.1 mm, the reactance value decreases and the no-load voltage increases by 1% [36]. To conduct a study considering the air gap clearance between the stator and permanent magnets, the deformation and vibration of the rotor should be taken into account during generator operation, and this will be considered in future work.

Meanwhile, in order to enable practical utilization of the currents and voltages induced by the proposed generator, a rectifier topology must be included in the circuit diagram as shown in Figure 8. Given that the induced currents and voltages contain harmonic content, an active rectifier, such as a six-switch rectifier [37], is recommended instead of a passive rectifier only to minimize high harmonic content of the currents and voltages. Thus, further research into the implementation of the active rectifier is also necessary.

As the weight of the generator is closely related to both mechanical losses and manufacturing costs, high power density enables the generator to be produced at a lower price than a generator with the same rated capacity. Additionally, it increases the efficiency of the generator. To confirm that the optimal design for the proposed generator was conducted reasonably, it was compared with other studies using the power generation/weight ratio, also known as power density (kW/kgf). To facilitate comparison with other generators, the weight of the proposed generator, excluding the bearing and cover, was determined to be 3.44 kgf. Table 7 provides the comparison of the proposed generator with other generators studied in several literatures. The power density increases as the rated capacity increases. For generators with a rated capacity greater than 8 kW, the power density exceeds 100 W/kgf. Some studies [10,14] have shown that the generator can achieve high power density even with a lower rated capacity. However, since the generator operates at a higher rotation speed, it needs a speed increaser gearbox to be installed, which ultimately increases the total weight of the wind turbine. The power density of the generator proposed in this study has been determined to be comparable to a 3 kW rated capacity [38,39] and to be higher than generators with similar rated capacities, such as [40,41]. Therefore, it is expected that the optimal design carried out in this study can contribute not only to reducing manufacturing costs and improving the mechanical efficiency for small-scale wind turbines, but also to increasing the power density even for wind turbines with a higher rated capacity.

**Table 7.** Comparison of the results for the generator with different rated capacity and power density.

| References | Rated Rotation Speed [rpm] | Rated Capacity [kW] | Power Density [kW/kgf] |
|---|---|---|---|
| Proposed in this study | 100 | 1 | 73.4 |
| [10] | 3000 | 0.39 | 130 |
| [42] | 100 | 0.4 | 48 |
| [40] | 129 | 1 | 62.5 |
| [41] | 300 | 2 | 59 |
| [38] | 200 | 3 | 75 |
| [39] | 300 | 3 | 85.7 |
| [14] * | 2000 | 3.6 | 600 |
| [4] | 100 | 7.5 | 51.4 |
| [15] | 210 | 8.1 | 143 |
| [43] | 100 | 12 | 109 |
| [44] * | 375 | 16.6 | 130 |
| [5] * | 211 | 20 | 178.6 |

* only consider weight of coppers and permanent magnets (active mass).

## 5. Conclusions

In this study, an optimal shape design of the axial-flux permanent magnet generator for 1 kW-class small-scale wind turbines was conducted to minimize the weight under the constraint of required active power. The electromagnetic analysis and sensitivity analysis for seven shape design parameters were carried out, which have an influence on the active power and weight of the generator. A response surface methodology was conducted based on the obtained results from the sensitivity analysis, and then the optimal shape design parameters of the generator were determined using the desirability function. The obtained results are as follows:

1.  A proposed outer-rotor type generator with axial-flux permanent magnets was suitable for a vertical-axis wind turbine, due to its ability to be vertically stacked in the center of a tower. The proposed generator was designed with a rated capacity of 1 kW, taking into consideration the sweep area of the wind turbine and its operation condition. Under the rated-load condition, each stage of the generator was required to generate 500 W with a torque of 48 Nm.
2.  The finite element method, based on the electromagnetic analysis, was used to estimate the electromagnetic characteristics of a half geometry of a single stage for the proposed generator. The induced currents and voltages obtained under the rated-load condition provided an active power of 199 W, which was corrected to consider a power factor of 0.8.
3.  Since the obtained active power was insufficient for the target specification, the optimal shape design was carried out to meet the target specification, including an active power of 250 W with weight minimization. The design space exploration was conducted using a sensitivity analysis to quantify variations in performance regarding changes in seven design parameters.
4.  The inner diameter, length of the stator and slot height were considered as the main shape parameters that influence both the active power and weight of the generator. Based on the results of the sensitivity analysis, a response surface model was constructed to establish the functional relationship between the weight, active power and design parameters.
5.  The optimal design parameters were identified using the desirability function based on the response surface model, resulting in an active power of 252 W and a weight of 5.04 kgf. The results indicate a 25% increase in active power while maintaining the weight of the initial model.

6. As a result of the comparison with other studies based on power density, the optimized generator provided a higher power generation/weight ratio for generators with a rated capacity of 3 kW. It is expected that an even higher power density can be achieved when the optimal design proposed in this study is applied to generators for wind turbines with higher rated capacities.

**Author Contributions:** All authors contributed to this work by collaboration. Conceptualization, H.J.P. and D.G.A.; methodology, H.J.P. and D.G.A.; software, H.J.P.; validation, H.J.P. and H.L.K.; formal analysis, H.J.P. and D.G.A.; investigation, H.L.K.; resources, H.L.K.; writing—original draft preparation, H.J.P.; writing—review and editing, S.H.H.; supervision, S.H.H.; project administration, S.H.H. All authors have read and agreed to the published version of the manuscript.

**Funding:** This research was funded by Dong-A University, Busan, Republic of Korea.

**Institutional Review Board Statement:** Not applicable.

**Informed Consent Statement:** Not applicable.

**Data Availability Statement:** The data presented in this study are available upon request from the corresponding author.

**Acknowledgments:** The authors are grateful for the financial support provided by research funds from Dong-A University.

**Conflicts of Interest:** The authors declare no conflict of interest.

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
