# Peer review of "Optimal Shape Design of Direct-Drive Permanent Magnet Generator for 1 kW-Class Wind Turbines"

_applsci, doi:10.3390/app13105856_

Round 1

Reviewer 1 Report

  1. In the introduction, you need to connect the state of the art to your paper goals. Please follow the literature review by a clear and concise state of the art analysis. This should clearly show the knowledge gaps identified and link them to your paper goals. Please reason both the novelty and the relevance of your paper goals.
  2. Better discuss industrial aspects of your research results in details.
  3. Conclusions must go deeper, it would be more interesting if the authors focus more on the significance of their findings regarding the importance of the interrelationship between the obtained results and the journal scope in the sector context, and the barriers to do it, what would be the consequences, in the real world, in changing the observed situation, what would be the ways, in the real world, to change/improve the observed situation.
  4. Some discussions are necessary for the introduction to provide the readers with a big picture. 
  1. Numerous minor mistakes in English writing have been found. Please polish the manuscript to avoid errors.
  2. The work is well written and provides good results, which are properly presented in the graphs, but their discussion can be deepened.
  3. The Abstract should be improved.
  4. From readers perspective, authors are suggested to incorporate detailed mathematical modelling of proposed framework.
  5. After inclusion of more detailed modeling and validation, authors are suggested to improve the conclusion of the paper based on findings and after impacts.
  6. Compare your results with the results of other researchers.
  7. The authors can consider the items below for improving the conclusion section: - Restate the research topic in conclusion. - Summarize the main points. - State the significance or results. - Avoid repeating information that you have already discussed. - Mention the model's name, and the advantages and disadvantages of the model. - Mention limitation of the study. - Provide some recommendations for future potential researchers.

Author Response

Thank you for considering my article for publication in applied sciences.  I am grateful to you and the reviewers for the valuable suggestions provided.
Here are responses to the reviewer comments through the attached file.

Reviewer 2 Report

Some comments and suggestions are given here:

- In the introduction, the objectives of the article should be clarified

- Interpretation of results needs to be improved.

- Iprove the quality of figure 11.

- I propose adding a list of the abbreviations.

Author Response

(The authors gave the same response as above.)

Reviewer 3 Report

The paper discusses optimal design of direct-drive permanent magnet generator for 1kW-class wind turbines. Wind generation is a topic demanded by practice. However, the paper has significant flaws.

The generator is connected directly to an active load. However, this type of generator does not provide the ways of controlling its modes. How can standard parameters such as voltage, frequency, allowed level of THD of the supply is obtained at various speed of wind?

Parameters of stator at cross-section area of AA’BB’ are shown in Fig. 4b. What is about such parameters in other cross-section areas located nearer to the rotational axes?

It must be shown that the used 2D model is applicable to the proposed design with its geometrical relations such as the outer and inner radius ratio, the dependance of sizes on the distance to the rotational axes.

In eq. (1), dD/dt is superfluous, quasistatic approximation is enough.

In eq. (2), use B instead of ( × ?).

Add the substitution B=( × ?).

Author Response

(The authors gave the same response as above.)

Round 2

Reviewer 3 Report

Most of my comments have been taken into account. For paper to be published, I thinck,  some words about the necessity of the converter must be highlighted. Also, the converter usually consists of a rectifier and an invertor. Some details on rectifier must be mensioned. Namely, it is assuned that it is active rectifier or at least with harmonic correction. The uncontrolled rectifier's current is significantly not sinusoidal. Also, rectifier is assumed to be tuned so as to behave like mere active resistor.
